# *HOXA10* and *HOXA11* in Human Endometrial Benign Disorders: Unraveling Molecular Pathways and Their Impact on Reproduction

**DOI:** 10.3390/biom15040563

**Published:** 2025-04-10

**Authors:** Lorin-Manuel Pîrlog, Andrada-Adelaida Pătrășcanu, Mara-Diana Ona, Andreea Cătană, Ioana Cristina Rotar

**Affiliations:** 1Department of Molecular Sciences, Faculty of Medicine, University of Medicine and Pharmacy “Iuliu Hațieganu”, 400012 Cluj-Napoca, Romania; lorin.pirlog@gmail.com (L.-M.P.); mara.ona17@gmail.com (M.-D.O.); 2Regional Laboratory Cluj-Napoca, Department of Medical Genetics, Regina Maria Health Network, 400363 Cluj-Napoca, Romania; 3Department of Oncogenetics, “Prof. Dr. I. Chiricuță” Institute of Oncology, 400015 Cluj-Napoca, Romania; 41st Department of Obstetrics and Gynecology, Faculty of Medicine, University of Medicine and Pharmacy “Iuliu Hațieganu”, 400006 Cluj-Napoca, Romania; cristina.rotar@umfcluj.ro

**Keywords:** cytokine signaling, decidualization, endometrial hyperplasia, endometrial polyps, endometrial receptivity, endometriosis, epigenetic regulation, extracellular matrix remodeling, fertility disorders, implantation failure

## Abstract

*HOX* genes, a family of conserved transcription factors, are critical for reproductive tract development and endometrial functionality. This review highlights the molecular underpinnings of *HOXA10/HOXA11* in reproductive health and their dysregulation in benign pathologies associated with infertility, such as endometriosis, adenomyosis, and endometrial polyps. These genes are dynamically regulated by estrogen and progesterone, with peak expression during the secretory phase of the menstrual cycle when implantation takes place. The molecular mechanisms underlying their action include the modulation of extracellular matrix (ECM) remodeling via metalloproteinases, cytokines like leukemia inhibitory factor, and cell adhesion molecules such as β3-integrin, all of which are essential for the differentiation of epithelial and stromal cells, as well as for trophoblast invasion. Aberrant *HOX* gene expression, driven by DNA hypermethylation or altered histone acetylation, compromises endometrial receptivity and implantation. For instance, reduced *HOXA10* expression in endometriosis stems from hypermethylation and chronic inflammation, disrupting immune modulation and cytokine signaling. Similarly, adenomyosis alters *HOXA11*-regulated ECM remodeling and β3-integrin expression, impairing embryo attachment. Furthermore, regulatory pathways involving vitamin D and retinoic acid offer promising therapeutic avenues pathways, as they enhance *HOXA10/HOXA11* expression and endometrial receptivity. This review underscores the critical molecular roles of *HOXA10/HOXA11* genes as biomarkers and therapeutic targets to optimize fertility outcomes and address reproductive pathologies.

## 1. Introduction

Genetic and genomic factors are key contributors to the development of various human diseases, including numerous endometrial disorders. The intricate molecular landscape governing mammalian reproduction is shaped by a complex interplay of genetic regulators, signaling pathways, and epigenetic modifications. Among these, homeobox (*HOX*) genes play a pivotal role in reproductive system development and function, particularly in establishing endometrial receptivity, implantation, and fertility [1].

This review aims to investigate the molecular interactions between benign reproductive system disorders, infertility, and the involvement of *HOXA10/A11* genes in their development and associated pathophysiology. By examining the regulatory networks influenced by these transcription factors, we seek to elucidate their contributions to common reproductive conditions, such as endometriosis, adenomyosis, and endometrial polyps. Understanding the molecular mechanisms underlying these pathologies is crucial, as they are leading causes of infertility and pregnancy complications [2].

Endometriosis is one of the most common benign gynecological conditions associated with infertility, affecting up to 50% of women with reproductive challenges [1,2]. The mechanisms linking endometriosis and infertility are multifactorial and include chronic inflammation, altered peritoneal immune environment, hormonal imbalances, particularly progesterone resistance, and anatomical distortions due to adhesions. These factors can impair folliculogenesis, ovulation, tubal function, and endometrial receptivity [1]. Moreover, molecular alterations in the eutopic endometrium of women with endometriosis, such as downregulation of *HOXA10/A11*, further compromise implantation by disrupting decidualization, immune cell function, and cytokine signaling [2,3]. As a result, even in the absence of gross anatomical abnormalities, as seen in other benign endometrial disorders like endometrial polyps and endometrial hyperplasia, endometriosis can significantly lower implantation and pregnancy rates, both naturally and during assisted reproductive technologies [1,3].

Although this review focuses primarily on *HOXA10/A11* gene dysfunction in reproductive disorders, it is important to acknowledge that these conditions arise from a multifaceted genetic landscape. Various other molecular pathways and genetic regulators have been implicated in endometrial disorders, contributing to their complexity. Our discussion of *HOX* gene abnormalities aims to provide a deeper understanding of their specific impact within this broader genetic framework, aligning with ongoing research efforts exploring key molecular and cellular mechanisms that govern human reproduction [3].

By synthesizing current knowledge on *HOXA10/A11* gene involvement in reproductive health and disease, this review contributes to the growing body of research that seeks to refine diagnostic tools, identify potential therapeutic targets, and improve fertility outcomes for individuals affected by these conditions.

## 2. Materials and Methods

### 2.1. Literature Search Strategy and Database Selection

A comprehensive literature search was conducted using PubMed, Google Scholar, and Scopus to identify relevant studies on the role of *HOX* genes in endometrial disorders, fertility, and associated molecular pathways. To ensure a comprehensive and systematic approach, we focused on key terms related to our study. These included *HOXA10* and *HOXA11* genes, as well as specific benign endometrial disorders such as endometriosis, adenomyosis, and endometrial polyposis. Given the relevance to reproductive health, we incorporated terms associated with fertility, implantation, pregnancy maintenance, and overall reproductive function. To explore the molecular mechanisms involved, we included molecular pathways, signaling pathways, immune modulation, cytokine interactions, and integrin-related processes. Additionally, we considered therapeutic aspects, integrating terms related to treatment, intervention, and therapy. Since inflammation plays a crucial role in these conditions, we also focused on immune response, cytokines, immunomodulation, regulatory factors, and gene regulation. This search strategy allowed us to retrieve a wide range of studies covering molecular mechanisms, immune interactions, and therapeutic interventions related to the function of *HOXA10* and *HOXA11* genes in reproductive health. By utilizing multiple databases, we ensured comprehensive coverage of peer-reviewed articles, citation networks, and the broader scientific literature, thereby enhancing the depth and reliability of our review.

### 2.2. Selection Criteria Overview

To ensure a systematic and high-quality selection of studies, specific inclusion and exclusion criteria were applied. These criteria were designed to focus on research exploring the role of *HOX* genes in endometrial disorders, fertility, and reproductive health, while filtering out studies that lacked relevance or scientific rigor.

Studies were included based on the following criteria:a.1.Studies exploring the role of *HOXA10/HOXA11* genes in human benign endometrial disorders, fertility, or reproductive health.a.2.Human studies were prioritized, though relevant animal studies were considered.a.3.Publication date between 1 January 2004 and 31 December 2024, with an emphasis on molecular pathways, genetic regulation, and clinical conditions such as endometrial polyposis, adenomyosis, endometriosis, and infertility.a.4.Both experimental and observational studies were included, along with systematic reviews, meta-analyses, and randomized controlled trials to ensure a comprehensive analysis of the available literature.a.5.English-language publications to ensure accurate interpretation.

Studies were excluded based on the following criteria:b.1.Studies that focused on non-*HOXA10/HOXA11* genes or non-reproductive health systems.b.2.Non-peer-reviewed articles, including opinion pieces and conference abstracts.b.3.Studies published before 1 January 2004.b.4.Animal studies were excluded from this review as our focus is solely on human reproductive health.b.5.Non-English publications due to potential translation and interpretation limitations.

### 2.3. Article Selection Process

To ensure a rigorous and systematic review of the literature, a comprehensive search was conducted across multiple databases, including PubMed, Scopus, and Google Scholar. The process of selecting relevant articles was guided by specific exclusion criteria to filter out studies that did not align with the research objectives. This process followed the PRISMA (Preferred Reporting Items for Systematic Reviews and Meta-Analyses) guidelines, as illustrated in the flowchart below.

Initially, a total of 678 records were identified from the selected databases. After removing 149 duplicate records, 529 studies were screened based on their titles and abstracts. At this stage, 353 reports were excluded for various reasons, such as lack of relevance to the research topic, insufficient methodological quality, or lack of sufficient data. The remaining 176 reports underwent full-text analysis to assess their eligibility. Following a more detailed review, an additional 65 reports were excluded, primarily due to methodological inconsistencies or irrelevance to the study’s scope. Ultimately, 111 studies were deemed suitable for inclusion in the final review.

The PRISMA flowchart (see Figure 1) visually represents the selection process, from the initial identification of studies to the final inclusion of eligible research articles. It outlines the number of records retrieved from different databases, the number of duplicates removed, and the filtering process through both abstract screening and full-text assessment. The exclusion reasons are categorized at each stage, providing transparency regarding the selection process. This structured approach ensures that only high-quality and relevant studies contribute to the review, enhancing the reliability of the findings.

## 3. Role of *HOX* Genes in Reproductive Health

Homeobox genes (*HOX* genes) are a group of highly conserved genes that control anterior–posterior (A-P) axial development. Their expression in the paramesonephric (Müllerian) duct respects a spatial order along the A-P axis and influences the development of the female reproductive system. From *HOXA9,* which controls the oviduct development, to *HOXA13,* involved in the formation of the ectocervix and upper vagina, each gene, when abnormally expressed, can lead to homeotic transformations or agenesia of certain regions.

Regarding reproductive health, *HOX* genes like *HOXA10* and *HOXA11* are essential to endometrial receptivity and implantation. They contribute to the structural and functional integrity of the endometrium, ensuring it is prepared for embryo implantation. Endometrial polyps (EPs), endometriosis, polycystic ovarian syndrome (PCOS), leiomyoma, and hydrosalpinx are associated with both the altered expression of *HOXA10* and *HOXA11* and infertility [1,2,3] (see Figure 2).

### 3.1. Benign Disorders of the Reproductive System

The failure to conceive after a year of consistent, unprotected intercourse is referred to as infertility. A couple can be infertile due to feminine factors (among them tubal injuries, PCOS, endometriosis, or adenomyosis), masculine factors, masculine and feminine factors, or idiopathic causes (infertility in the absence of a demonstrable factor). Implantation may also be hampered by uterine anomalies such as EPs, fibroids, and congenital deformities. Infertility is also exacerbated by ovulatory abnormalities, decreased ovarian reserve, and damage to the fallopian tube from infections or pelvic inflammatory disease. The primary uterine disorders that are closely linked to the dysregulation of *HOXA* gene expression will be covered in more detail.

PCOS is a prevalent endocrine disorder that affects the reproductive system. It is characterized by the triad of hyperandrogenism, oligoovulation, or anovulation and the presence of polycystic ovaries, although not all three criteria are required for diagnosis [4,5,6]. The prevalence ranges from 5 to 18%, with many cases still undiagnosed [7].

Endometriosis is a chronic inflammatory disorder that implies the existence of endometrial tissue outside the uterine cavity. Endometriosis sites can be identified in the peritoneum, pelvic ligaments, ovaries, or even the bowel and bladder. Of the symptomatic patients, most complain of pain (presenting as chronic pelvic pain, dyspareunia, or dysmenorrhea) or infertility. They can be accompanied by urinary or digestive symptoms.

Adenomyosis is a benign pathology in which endometrial tissue grows within the myometrium, often resulting in heavy menstrual bleeding, chronic pelvic pain, and reduced fertility. The severity of symptoms is related to the extent of myometrial involvement and often correlates with the patient’s age [8,9].

EPs are focal proliferations of endometrial glands and stroma organized around a vascular core. They arise from the endometrial surface and project into the uterine cavity [10]. As many as 32% of uterine polyps are found on the posterior uterine wall, with the anterior and lateral walls being less frequently affected [11]. They can be single or multiple, with different sizes, ranging from 5 mm to a few centimeters. Based on their morphology, they are either pedunculated or sessile. Most of them are benign and express estrogen receptors [12]. The exact prevalence of EPs is unknown, as most of the cases are asymptomatic. Prevalence rises with age, less than 1% of young women are affected, while the rate increases to 9.2% in women over 30. Polyps are more common in contraceptive users, and their prevalence reaches 25% in women undergoing hormone therapy, possibly due to the combined effects of age and hormonal treatment [13].

EPs are the consequence of hormonal imbalances, genetic alterations, and environmental factors such as inflammation. The glandular cells in EPs have a high concentration of estrogen receptors (ERs) and a lower expression of progesterone receptors (PRs) compared to the rest of the endometrium, which could explain why estrogen stimulation is the key factor blamed for their pathogenesis. In contrast, stromal cells display low receptor expression, rendering them less responsive to the cyclic changes in the normal endometrium. Tamoxifen, a selective estrogen receptor modulator (SERM), stimulates the estrogen receptor in the uterus, which is associated with an increased risk of developing polyps. Hormonal replacement therapy and the use of oral contraceptives also lead to a higher risk of EPs [14,15].

Elevated levels of aromatase increase the conversion of androgens to estrogens in the adipose tissue; therefore, obesity is considered a risk factor for EPs.

De Spiezio classification of uterine polyps is a classification of utmost importance, as it plays a key role in assessing the risk of malignancy and guiding the therapy.

Localized chronic inflammation, driven by mast cell secretion of cytokines and growth factors, along with elevated cyclooxygenase-2 levels, is implicated in EPs formation through the promotion of angiogenesis and tissue growth.

For EPs to form, there must be an imbalance between mitotic activity and apoptosis that disrupts normal endometrial development. Bcl-2, an apoptosis inhibitor, shows increased expression in proliferative-phase polyps, contributing to their persistence, while apoptosis markers like DNA fragmentation factors (DFF40 and DFF45) are reduced. Ki-67, a proliferation marker, is notably elevated in tamoxifen-treated women. Additionally, cytogenetic studies have identified chromosomal rearrangements in stromal cells, particularly in regions 6p21–22, 12q13–15, and 7q22, as key contributors to endometrial polyp formation [14].

### 3.2. Relationship Between Benign Endometrial Conditions and Fertility

PCOS is a major cause of female infertility. The key factors that lead to this are hormonal imbalances, with high levels of androgens and luteinizing hormone, follicular dysfunction that leads to anovulation, insulin resistance, chronic low-grade inflammation, endometrial abnormalities caused by unopposed estrogen exposure, and lifestyle factors. Even after resolving anovulation, achieving a successful pregnancy remains challenging, with miscarriage occurring in 30–50% of all conceptions [3,16].

Endometriosis affects almost 10% of women of reproductive age, with one-third experiencing fertility challenges. The mechanisms by which endometriosis causes infertility are still unclear. Chronic inflammation can lead to an imbalance in local immune responses, as well as adhesions and fibrosis, all of which in turn affect implantation. Endometrial receptivity is altered by both hormonal dysregulation—such as resistance to progesterone and estrogen prevalence—and persistent inflammation. Ovarian function may also be affected [17].

Adenomyosis can contribute to infertility in many ways; it disrupts endometrial receptivity, alters the uterine contractility, and distorts the uterine cavity. The abnormal myometrial contractions, chronic inflammation, and the modified expression of key molecules like *HOXA10*, β3-integrin, and osteopontin make sperm migration, oocyte transport, and embryo implantation difficult. These changes reduce implantation rates, increase the risk of miscarriage, and negatively affect in vitro fertilization outcomes despite good embryo quality [18].

EPs are the most common reported intrauterine structural abnormality that leads to infertility and have been found in 15% to 24% of women with infertility [11,19]. The chief complaint of women with EPs is AUB (heavy menstrual bleeding, intermenstrual bleeding, and postmenopausal bleeding). This is caused by the congestion of the polyps’ stroma that leads to venous stasis and apical necrosis [14,20]. Most symptomatic polyps contain mixed epithelial components, and interestingly, factors such as the size, location, and number of polyps do not appear to correlate with the extent of the bleeding [17]. EPs are usually asymptomatic and are often detected during a pelvic ultrasound. In infertile women, due to their known association with infertility, EPs should be removed before proceeding with further fertility treatments.

While the exact mechanism by which they contribute to infertility is still unknown, a possible explanation is their capacity to mechanically obstruct sperm transportation and hinder the implantation of the embryo [19]. Proof of this theory resides in the improvement in in vitro fertilization results after the surgical treatment of the polyp. EPs located at the uterotubal junction are associated with a significantly higher pregnancy rate after excision (57.4%) compared to those on the posterior (28.5%), lateral (18.8%), and anterior (14.8%) uterine walls [11]. However, the fertility outcomes after polypectomy are not dependent on the size of the polyp, which weakens this argument [21].

Another factor that could cause infertility was glycodelin. Glycodelin is a glycoprotein found in the endometrial epithelium that plays a key role in regulating fertility by inhibiting sperm–oocyte binding and natural killer (NK) cell activity. Its levels vary during the menstrual cycle, reaching the lowest point around the moment of ovulation to aid the fertilization process, and high levels after ovulation, protecting the process of implantation by decreasing NK cell activity. In patients with EPs, glycodelin levels are abnormally high, especially during the pre-ovulatory phase, which may interfere with fertilization and implantation by disrupting normal endometrial receptivity [9,11]. The expression of *HOXA10* and *HOXA11* genes is altered in patients with polyps which may compromise the endometrial receptivity [14,19].

## 4. Overview of *HOX* Genes and Endometrial Function

### 4.1. Overview of HOX Genes and Their Roles in Cellular Differentiation and Spatial Development

The *HOX* gene family is a highly conserved transcription factor family with a crucial role for embryonic development and establishing body plans in bilaterian organisms. They belong to a subgroup of homeobox genes and are arranged on chromosomes in clusters referred to as *HOX* clusters. Multiple *HOX* genes are found in each cluster, their expression being spatially and temporally controlled to ensure that tissues and organs develop adequately along the anterior–posterior (head-to-tail) axis. Thirty-nine functional *HOX* genes are dispersed widely throughout human *HOX* clusters. These are further separated into 13 paralogous groups according to positional homology and sequence similarity among the clusters [22].

The homeobox, a highly conserved 180-base-pair sequence found in *HOX* genes, encodes the homeodomain, a 60-amino-acid DNA-binding protein region. Regarding the human body, four distinct clusters: *HOXA*, *HOXB*, *HOXC*, and *HOXD*, are found on the chromosomes 7p15.2, 17q21.32, 12q13.13, and 2q31.1. Ancestral gene duplication events are hypothesized to be the source of the mentioned clusters [23,24].

Positional identity throughout embryogenesis is the main function of *HOX* genes. They affect the expression of downstream target genes that govern organogenesis, migration, and cell differentiation. The concept of collinearity governs the pattern of *HOX* gene expression. While temporal collinearity oversees the genes at the 3′ end of a cluster being activated earlier in development than those at the 5′ end, spatial collinearity regulates that the genes at the 3′ end of a cluster are expressed in anterior regions of the body, while those at the 5′ end are expressed in posterior regions. Moreover, another kind of collinearity was noted: quantitative collinearity. When multiple *Hox* genes are activated at a specific point along the embryo’s anterior–posterior axis, the expression of the cluster’s most posterior gene (5′) is stronger than that of the more anterior genes (3′). In humans, quantitative collinearity may not be as noticeable or significant. The existence and importance of quantitative collinearity in the regulation of the human *HOX* gene may be further clarified by additional research in transcriptomics and chromatin structure [25,26] (see Figure 3).

The optimal segmentation of the spinal column and the development of limbs depend on *HOX* genes. Their essential function in evolution is amply demonstrated by the fact that they are conserved across species. Comparative research shows that evolutionary variations in body morphology can result from *HOX* gene mutations or misexpression, which brings more insight into the variety of body plans found in different animals. For instance, it is believed that variations in the expression patterns of the *HOX* gene possess an essential part in the differentiation of the nervous system and the differences in the development of the limbs between vertebrates and arthropods. Therefore, dysregulation of the *HOX* gene is strongly associated with a diversity of human disorders, such as malignancies, notably leukemia, breast and lung cancer, and congenital abnormalities, particularly limb malformations [27,28,29].

### 4.2. Specific Focus on HOXA10 and HOXA11 Genes

*HOX10* and *HOX11* genes belong to the broader *HOX* gene family, which orchestrates embryonic development and guarantees appropriate body patterning. They possess vital roles in skeletal, genitourinary, and hematopoietic system development and are involved in tumoral growth, when an aberrant expression is present. Importantly, *HOXA10* and *HOXA11* are maximally expressed in the endometrium [30]. These genes are paralogous, sharing evolutionary ancestry and sequence homology, located within the four *HOX* gene clusters. *HOX10* genes are positioned posteriorly within their respective clusters and play critical roles in defining the axial and appendicular skeleton, whereas *HOX11* genes are positioned adjacent to *HOX10* genes and are involved in specific developmental processes, particularly in the kidneys and forelimbs [31].

The sequence-specific transcription factor *HOXA10* attaches itself to the DNA sequence 5-AA[AT]TTTTATTAC-3 on the 5′ regulatory region. The 410-amino-acid protein that the human *HOXA10* gene codes for possesses a DNA-binding domain that encompasses amino acids 336–395 and has an approximate molecular mass of 42 kDa. Histone deacetylase activity and DNA-binding transcription factors are linked to the primary molecular functions of *HOXA10*, while anterior/posterior segment specification is linked to vital biological activities. Multiple phosphorylation, methylation, acetylation, and ubiquitination sites are present in human *HOXA10*. There have also been recent reports of the sumoylation of *HOXA10*, which inhibits *HOXA10* protein stability and transcriptional activity without affecting its subcellular localization [32].

*HOX10* genes include *HOXA10*, *HOXB10*, and *HOXD10*, having various roles in the organism. They are responsible for skeletal development, regulating the patterning of the vertebral column, particularly the lumbar region, by suppressing rib formation. Their overexpression is associated with ovarian and breast cancers, due to their role in regulating cell proliferation and differentiation. Regarding the reproductive system, *HOXA10* plays a vital role in uterine development and endometrial receptivity, which is crucial for implantation, as aberrant *HOXA10* expression is linked to infertility and endometriosis. Moreover, they possess a central role in the segmentation of the female reproductive tract specifically to demarcate the boundary of the fallopian tube and the anterior uterus [23,30].

*HOX11* genes include *HOXA11*, *HOXB11*, and *HOXC11*. Similarly to *HOX10* genes, they are involved in the production of the normal skeleton, especially the development of the radius and ulna. They are necessary for the development of the metanephric kidney, with *HOXA11* and *HOXD11* regulating nephron differentiation and renal morphogenesis, and their mutations possibly resulting in congenital anomalies of the kidney and urinary tract [33].

Expressed during T-cell development, *HOX11*, also named *TLX1*, functions as an oncogene by altering cell cycle regulation and apoptosis pathways, causing lymphoblastic leukemia (T-ALL) [34]. While *HOXA11* has been extensively studied in uterine biology, *HOXC11* has limited documented roles in the reproductive system. However, as a member of the *HOX11* paralogous group, it is hypothesized to contribute to cellular differentiation in the reproductive tract [35].

### 4.3. Regulation of Stromal Cell Differentiation by HOX Genes

*HOX10* and *HOX11* genes have a key function in monitoring and guiding endometrial receptivity to achieve an effective embryo implantation. They affect the expression of factors that regulate the uterine lining’s receptivity, such as cell adhesion molecules, cytokines, and growth factors. Thereby, in case of dysregulation conditions like infertility, recurrent implantation failure and suboptimal outcomes in assisted reproductive technologies occur, prioritizing their relevance in reproductive health.

The processes of stromal differentiation and decidualization, which are critical for implantation, are the main functions of *HOX10*. It modulates the synthesis of matrix metalloproteinases (MMPs), integrins (including integrin β3), and cytokines such as leukemia inhibitory factor (LIF). These elements promote trophoblast invasion, improve extracellular matrix (ECM) remodeling, and strengthen cell adhesion. On the other hand, *HOX11* promotes immunological regulation and glandular epithelial growth. It controls genes that are essential for the maturation of decidual stromal cells, such as prolactin (PRL) and insulin-like growth factor-binding protein 1 (IGFBP1). Additionally, it promotes the release of immuno-modulatory chemicals, thereby providing an immune environment that is advantageous for the implantation of embryos [31,36,37].

### 4.4. Regulation of Epithelial Cell Differentiation by HOX Genes

In addition to their roles in stromal cell decidualization, *HOXA10* and *HOXA11* also influence the endometrial epithelium through distinct molecular mechanisms that differ fundamentally from their effects on the stroma. In the endometrial epithelium, these genes primarily regulate cell adhesion molecules (e.g., integrin β3) and cytokine signaling pathways (e.g., LIF, IL-6), which are essential for establishing a receptive epithelial surface for embryo attachment [9,11,36,37,38]. The modulation of integrin expression and cytokine activity in epithelial cells enhances cell-to-cell adhesion and communication, promoting a structurally and biochemically conducive environment for implantation [19].

In contrast, the stromal compartment undergoes decidualization, a differentiation process initiated by progesterone signaling and regulated by *HOXA10* and *HOXA11*. This process involves extensive remodeling of the extracellular matrix (ECM), upregulation of decidual markers PRL and IGFBP1, and the establishment of a pro-implantation immune environment [36,37]. Unlike epithelial cells, which primarily respond through modulation of adhesion molecules and cytokines, stromal cells undergo profound structural and functional changes to support implantation and early pregnancy [36,37,38].

Altered expression of *HOX* genes in the epithelium can result in dysregulated expression of mucins and adhesion molecules, thereby compromising the initial steps of implantation. Conversely, disruptions in stromal decidualization affect ECM remodeling, cytokine production, and overall tissue receptivity. These differential responses underline the specialized functions of epithelial and stromal compartments, with each contributing uniquely to endometrial receptivity through distinct molecular pathways. However, their interplay is crucial, as epithelial receptivity facilitates the initial attachment of the embryo, while stromal decidualization provides the structural and immunological support required for successful implantation [11,36,37,38].

## 5. *HOX* Gene Expression During the Menstrual Cycle and Implantation

### 5.1. Regulation of HOXA Gene Expression

Relaxin, retinoic acid (RA), sex hormones, and 1,25-dihydroxycholecalciferol-3 are known to control the expression of *HOXA* genes. The susceptibility of *HOX* genes to RA varies according to the cluster position. Because of their high RA sensitivity, genes at the 3′ end require lower doses to modulate their expression; meanwhile, sex steroids have a stronger effect on genes at the 5′ end [31].

### 5.2. Role of HOXA10 and HOXA11 in Uterine Function and the Menstrual Cycle

Both embryonic differentiation and posterior patterning depend on estrogen *HOX* signaling. Estradiol (E2) and progesterone (P4) control the expression of *HOXA10* and *HOXA11* to guide uterine growth, differentiation, and receptivity. Their expression fluctuates during the menstrual cycle, with E2 and P4 working individually and in concert, which was demonstrated to have an additive effect.

*HOXA10* and *HOXA11* exhibit dynamic expression patterns throughout the menstrual cycle, aligning with hormonal changes that prepare the endometrium for implantation. During the proliferative phase, their expression levels are relatively low. As E2 levels increase, they induce the growth and proliferation of the endometrial lining, initiating a mild upregulation of *HOXA10*. The combined effects of E2 and LH prime the endometrial environment, leading to a significant increase in *HOXA10* expression throughout ovulation. The uterus is thereafter prepared for the progesterone-driven changes that take place during the secretory phase.

In this phase, P4 from the corpus luteum becomes the prevailing hormone, working synergistically with E2. *HOXA10* and *HOXA11* reach their peak expression levels during the mid-to-late secretory phase, which coincides with the period of uterine receptivity, the so-called window of implantation or window of receptivity. *HOXA10* plays a pivotal role in endometrial differentiation and receptivity by promoting stromal and epithelial cell changes, such as decidualization. *HOXA11* also supports decidualization and improves uterine receptivity by regulating ECM remodeling and stromal cell adhesion (see Table 1) (see Figure 4) [31,38].

### 5.3. Expression Patterns in the Presence or Absence of Pregnancy

If the establishment of pregnancy does not occur, the regression of the corpus luteum, which becomes corpus albicans, leads to a decline in P4 and E2 levels. This triggers a downregulation of *HOXA10* and *HOXA11*, contributing to the shedding of the functional endometrial layer. Whenever an embryo is present, corpus luteus activity persists due to its stimulation by human chorionic gonadotropin synthetized by trophoblastic cells, which will increase the expression of *HOXA10* in the endometrial stromal cells, ensuring optimal conditions for embryonic growth [31,39].

*HOXA10* and *HOXA11* play a cardinal role in implantation by converting endometrial stromal cells into specialized decidual cells, as well as creating the structural and physiological conditions required to facilitate the embryo’s attachment and invasion. To acquire the decidual phenotype, the release of several proteins is required, with the main ones being PRL and insulin-like growth factor-binding protein-1 (IGFBP-1) [31,39] (see Figure 5).

### 5.4. HOXA Genes in Lymphocyte Function and Immune Response

*HOXA10* and *HOXA11* genes may affect lymphocyte function by regulating the expression of cell surface markers and cytokines. These genes are involved in various signaling pathways related to immune responses, including integrins, cytokines such as IL-6, and adhesion molecules that control the migration and interaction of immune cells [31,36,40]. For instance, *HOXA10*’s role in cytokine regulation is believed to impact immune cell function during pregnancy. While the specific lymphocyte markers regulated by *HOX* genes remain largely unstudied, it is possible that their dysregulation could contribute to inflammatory responses and implantation failure. Additionally, it is important to note that *HOX11* is expressed during T-cell development and functions as an oncogene, disrupting cell cycle regulation and apoptosis pathways, which can lead to T-cell acute lymphoblastic leukemia [34]. Further research is needed to identify the specific lymphocyte molecules influenced by *HOXA* genes and their role in immune cell trafficking during implantation (see Figure 5).

### 5.5. HOXA11’s Role in ECM Remodeling and Immune Regulation

*HOXA11* is particularly involved in remodeling the ECM by regulating the MMPs, which split components of the ECM to assist the very first step of implantation, the trophoblast invasion, enabling the embryo to pervade the endometrial lining. In this regard, *HOXA10* also modulates the immune response, especially in early pregnancy, by controlling the expression of several cytokines such as LIF, interleukin-6 (IL-6), transforming growth factor-beta (TGF-β), tumor necrosis factor-alpha (TNF-α), and Vascular Endothelial Growth Factor (VEGF) (see Figure 5) [40].

### 5.6. Connection Between HOXA Genes and OVGP1

A connection between the two *HOXA* genes engaged in implantation and OVGP1 (Oviductal glycoprotein 1), also known as oviductin, has been observed. Studies suggest that *HOXA10* can upregulate the production of OVGP1 in the endometrium. Progesterone controls the expression of OVGP1 and *HOXA* genes. The endometrium becomes appropriate for embryo nidation during the so-called “window of implantation”, which appears after hormonal effects during the secretory phase of the menstrual cycle [30].

To prepare the endometrium for implantation, P4-induced genes, *HOXA10* and *HOXA11*, have elevated expression levels during this phase, whereas OVGP1 is responsive to P4, as it controls its expression during this phase [41]. Several downstream genes involved in reproductive processes, specifically those influencing ECM remodeling, cell adhesion, and immunological responses, all of which have significance for successful implantation, are regulated by *HOXA10* and *HOXA11*. As part of a broader network of gene expression indispensable for appropriate implantation, these *HOX* genes may indirectly influence OVGP1 (see Figure 4 and Figure 5) [42].

### 5.7. Vitamin D and HOXA Gene Expression

In addition, vitamin D can increase the expression of *HOXA10* and *HOXA11* by activating the vitamin D receptor (VDR) in a variety of tissues, including the endometrium. Vitamin D is crucial for fertility and reproductive health, as a great number of studies indicate that it might act together with other hormones, such as P4 and E2, to control gene expression during the secretory phase of the menstrual cycle. This could contribute to higher uterine receptivity and set up the endometrium for embryo implantation [31,43].

### 5.8. Glycodelin and HOX Genes in Implantation

Glycodelin, a glycoprotein found in the endometrial epithelium, plays a critical role in regulating fertility by inhibiting sperm–oocyte binding and modulating NK cell activity. Its levels fluctuate throughout the menstrual cycle, with the lowest levels around ovulation to facilitate fertilization, and higher levels after ovulation to protect implantation by reducing NK cell activity. In patients with EPs, glycodelin levels are abnormally elevated, particularly during the pre-ovulatory phase, which may disrupt normal endometrial receptivity, hindering fertilization and implantation. Additionally, altered expression of the *HOXA10* and *HOXA11* genes, commonly observed in patients with EPs, can compromise endometrial receptivity and contribute to implantation failure. NK cells, essential in the early stages of pregnancy, recognize and respond to changes in the endometrial cells during implantation. Dysregulation of *HOXA10* and *HOXA11* expression may affect NK cell function, potentially leading to implantation failure. Glycodelin, which is modulated by these *HOX* genes, regulates NK cell activity by inhibiting their cytotoxic responses. While direct studies on the modulation of NK cell receptors by *HOX* genes are limited, it is hypothesized that these genes influence the immune landscape of the endometrium, promoting a favorable environment for embryo implantation. Future research could focus on examining the direct impact of *HOX* gene expression on NK cell markers and cytotoxic activity in uterine diseases like endometriosis and adenomyosis [9,11,14,19] (see Figure 5).

## 6. Disrupted *HOX* Gene Expression in Endometrial Benign Disorders

### 6.1. HOXA10 Expression in Gynecological Conditions

Numerous gynecological conditions that affect the uterus, including adenomyosis, polyps, fibroids, and leiomyomas, have changed *HOXA10* expression. *HOXA10* expression is decreased in stromal cells and endometrial tissue in patients with adenomyosis, which hinders decidualization in vitro. Reduced *HOXA10* expression is also seen in animal models of Tamoxifen-induced adenomyosis, which may be brought on by overexpression of HIF-2α, dysregulation of IL-10, or IL-33/p-STAT3 signaling. Although more research is required, this reduction may disrupt decidualization and lead to infertility. EPs and leiomyomas have also been found to exhibit similar decreases in *HOXA10*, which are frequently associated with hypermethylation of the *HOXA10* promoter [44,45]. It is still unknown, though, if this reduction is a result of these conditions or their cause.

### 6.2. Role of HOXA10 in Tissue Homeostasis and Differentiation

Although there are few experimental data connecting *HOXA10* to uterine abnormalities, research indicates it is important for controlling tissue homeostasis and differentiation. *HOXA10* is a member of the *HOX* family, which is essential for angiogenesis, apoptosis, differentiation, and cell growth [28,46,47]. Gynecological cancers, such as breast and ovarian tumors, have been linked to the abnormal transcription of *HOX* genes. When dysregulated, DNA methylation, a crucial epigenetic regulator of *HOX* gene expression, plays a role in *HOX*-related illnesses, such as cancer [48]. *HOXA5* affects apoptosis, differentiation, and cell proliferation in cancer [49,50], whereas *HOXB9* expression is associated with immune response and prognosis in a variety of cancers [51]. Better tumor diagnosis and treatment may result from an understanding of the function of *HOX* genes in development and disease progression [28,52]. Future studies should investigate whether decreased *HOXA10* contributes to infertility and uterine disorders.

It is important to distinguish between the regulation of *HOX* genes at the DNA level and the biological activity of their protein products. DNA methylation in the promoter region of *HOX* genes regulates gene transcription by controlling the accessibility of transcriptional machinery, leading to increased or decreased mRNA synthesis. This ultimately affects the amount of HOX protein produced, which in turn mediates downstream effects such as modulation of cell proliferation, differentiation, and apoptosis. However, it is the translated HOX proteins, acting as transcription factors, that directly interact with target gene promoters to regulate developmental and pathological processes. In addition, post-transcriptional mechanisms (e.g., alternative splicing) and post-translational modifications can further influence the function, stability, and localization of HOX proteins, adding an additional layer of complexity to their regulation in both normal physiology and disease [28,46,47,48,50].

### 6.3. HOXA10 and Endometrial Hyperplasia

There have been conflicting findings about *HOXA10* expression in endometrial hyperplasia, which is defined by aberrant endometrial tissue proliferation. While the differences were not statistically significant, one study found that *HOXA10* levels gradually decreased across simple, complex, and atypical hyperplasia. *HOXA10* expression in normal, hyperplastic, and FIGO nuclear grade 1 endometrial tissues did not differ significantly, according to another study [53].

However, experimental evidence supports a connection more strongly. Endometrial hyperplasia resulted from increased proliferation of epithelial cells in mice with decreased *HOXA10* expression via *Hoxa10* shRNA. Like women with hyperplasia, these mice also showed altered expression of β-catenin, *SOX9*, *YAP1*, and estrogen and progesterone receptors [53,54]. Furthermore, human endometrial cancer cell lines that overexpressed *HOXA10* showed decreased cell proliferation, indicating that *HOXA10* regulates the growth of epithelial cells. Although there are few clinical data connecting *HOXA10* to endometrial hyperplasia, experimental research clearly shows that it plays a role in the development of hyperplasia. The overall evidence supporting *HOXA10*’s involvement in this condition is deemed moderate due to the limitations of both human and animal models [44].

### 6.4. HOXA10 in Endometriosis

Women with endometriosis exhibit significantly lower levels of *HOXA10* gene expression than women without the condition; this pattern is seen in various populations. Fertility issues such as primary infertility, implantation failure, and decreased endometrial receptivity are closely linked to this decline. Several types of endometriosis, including superficial peritoneal (SE), ovarian (OE), and deep infiltrating endometriosis (DE), have lower *HOXA10* expression in both eutopic and ectopic endometrial tissues [55,56]. Although this conclusion is supported by most research, some studies present contradictory findings about *HOXA10* expression in ectopic lesions, with some demonstrating decreased expression and others showing increased expression. According to a review, *HOXA10* may play a role in endometrial cells being displaced close to Müllerian structures because of endometriotic lesions. This could lead to implantation failure, which is a major cause of infertility in patients with endometriosis [55,57].

Distinct endometriosis subtypes exhibit different levels of *HOXA10* expression. Compared to women with DE or OE, those with SE have lower antral follicle counts, longer infertility durations, and more alterations in *HOXA10* expression. SE is more likely to experience implantation failure, whereas DE and OE are less affected. Furthermore, lower implantation success rates are associated with lower *HOXA10* expression in adenomyosis [55].

Research indicates that there is a correlation between increased *HOXA10* expression and low *HOXA10* DNA methylation levels in ectopic endometrial tissues. Increased stromal cell activity surrounding the endometrial glands in ectopic tissues, which resembles the eutopic endometrium [58], could be the cause of this. However, by interfering with the *HOX* “code” [59], decreased *HOXA10* expression in endometrial stromal cells may lead to the formation of ectopic tissue. Research on *HOXA10*’s function in endometriosis is necessary, though, as the mechanisms underlying the development of ectopic endometrium are still poorly understood.

Women with endometriosis have methylated *HOXA10* in their endometrium during the secretory phase; methylation patterns are similar in all populations. The most researched location for DNA methylation is the *HOXA10* promoter region. To fully comprehend *HOXA10*’s function in endometriosis and its potential as a biomarker for diagnosis and prognosis, more research is necessary. Determining the levels of *HOXA10* methylation is essential for creating targeted DNA demethylation treatments for endometriosis [60].

### 6.5. HOXA10/HOXA11 and Endometrial Polyps

Infertility, AUB, and an elevated risk of endometrial cancer are all linked to EPs, a common benign gynecological condition. Although their exact cause is unknown, they are associated with chronic inflammation and hormonal imbalances (such as hyperestrogenism with low/absent progesterone due to anovulation in PCOS, obesity, and gonadotropin therapy). Age, tamoxifen use, obesity, and hypertension are risk factors [61]. EPs can include either immature or functional endometrium that is insensitive to hormonal stimuli and frequently manifests as cystic hyperplasia during the menstrual cycle [62].

Through molecular disruptions and mechanical obstruction that affect sperm transport and embryo implantation, EPs reduce fertility. These disturbances include decreased expression of *HOXA10* and *HOXA11*, which are important indicators of endometrial receptivity [63], as well as increased levels of glycodelin, aromatase, and inflammatory markers. Their development is further influenced by cytogenetic abnormalities in regions like 6p21–p22, 12q13–q15, and 7q22 [61].

Particularly in premenopausal women, EPs are commonly linked to chronic endometritis (CE), indicating a possible pathological continuum. EP tissues are frequently found to contain CD-138 plasma cells, which are a sign of CE [64]. Chronic inflammation disrupts endometrial receptivity by changing the expression of genes related to cytokine activation, apoptosis, and proliferation. Infertility and implantation failure are associated with this inflammation and decreased expression of *HOXA10* and *HOXA11* [62,65,66].

There is ongoing debate regarding the relationship between CE and EPs. According to certain research, macropolyps (EPs) and micropolyps have different etiologies. Micropolyps are thought to arise in inflammatory environments, whereas macropolyps are thought to form in environments that are sensitive to estrogen [67,68]. Some EP samples lack plasma cell infiltration, a characteristic of CE, which calls into question a causal relationship [69]. Distinct pathophysiological processes are also suggested by the differences in growth factor expression between micropolyps and EPs [68].

According to recent research, single and multiple EPs might have different origins. While single EPs have little association with inflammation [70], multiple EPs are more commonly linked to endometritis. This variation highlights the intricacy of EP development and calls for specialized approaches to diagnosis and treatment.

The possibility of recurrence and cancer in EPs emphasizes the need for a better understanding of their pathophysiology. Recent studies suggest that inflammatory, genetic, and hormonal factors interact in a complex way. Polyps in other mucosal tissues, such as the urinary, respiratory, and digestive tracts [71,72], may be caused by similar mechanisms. The generalizability of observational studies that concentrate on specific populations, such as those with AUB or infertility, is limited [68,73,74,75].

## 7. Interplay Between NF-κB and *HOX* Genes

Inflammation is a fundamental biological process that underlies a wide range of physiological and pathological conditions, from immune responses to chronic diseases. Given its central role in disease progression, understanding the molecular players that regulate inflammation is crucial. NF-κB is a key transcription factor involved in innate immunity, inflammation, and embryonic development, while *HOX* genes, particularly *HOXA10* and *HOXA11*, have been implicated in immune regulation and tissue homeostasis. Since both NF-κB and *HOX* genes contribute to inflammatory processes and disease pathogenesis, we aimed to explore their interplay, highlighting how their regulatory mechanisms influence immune responses, development, and potential therapeutic strategies.

### 7.1. NF-κB Pathway Overview

Like *HOX* genes, NF-κB is essential for innate immunity, adult inflammation, and embryonic development. Although NF-κB was first identified as being involved in inflammation, apoptosis, and proliferation, it has since been found to be crucial for developmental processes. It is crucial role in vertebrate limb bud outgrowth, proliferation, positional identity [76,77,78], and dorsoventral patterning was shown in studies conducted in 1998. Furthermore, NF-κB parallels the functions of *HOX* genes to regulate processes such as hematopoietic stem cell fate and mammary duct proliferation [76,79].

RelA (p65), c-Rel, RelB, NF-κB1 (p50), and NF-κB2 (p52) are members of the NF-κB family of transcription factors, which was identified in 1986 by David Baltimore’s group. The most prevalent of these proteins’ homo- or heterodimeric complexes is the p65/p50 heterodimer [80]. There are two primary ways that NF-κB functions: canonical and non-canonical. LPS, TNF-α, cytokines, and mitogens all activate the canonical pathway [80,81]. IKKs phosphorylate and break down IκB-α, allowing NF-κB to move to the nucleus after IκB-α sequesters NF-κB in the cytoplasm. To resolve inflammation, it then self-regulates via NFKBIA, the gene encoding IκB-α, and controls genes involved in adhesion molecule expression, cytokine and chemokine production, and anti-apoptosis [76,82].

LT, RANKL, CD40L, and BAFF-F.56 are ligands that activate the non-canonical pathway. After p100 is phosphorylated by IKKα, it is processed into p52, which dimerizes with RelB to activate the genes necessary for the development of secondary lymphoid organs and immune responses [83]. Target specificity may be guided by chromatin remodeling proteins, even though they share NF-κB binding sites. Interestingly, certain ligands, like CD40L and RANKL, can activate both pathways [76] (see Figure 6).

### 7.2. NF-κB and HOX Genes: Crosstalk in Immunity and Development

These revelations highlight the diverse functions of NF-κB in immunity, inflammation, and development, and they may have therapeutic ramifications for illnesses involving this pathway.

There is proof that the NF-κB pathway and *HOX* genes play interrelated roles in immune regulation and embryonic development. While NF-κB is necessary for dorsoventral patterning, notochord development, hematopoiesis, and organ formation, *HOX* genes are crucial for body patterning and positional identity [84]. Dorsoventral patterning is mediated by Drosophila homologs of IκB-α (cactus) and c-Rel (dorsal), while anteroposterior patterning is regulated by *HOX* transcription factors, indicating shared regulatory mechanisms [76].

A study found that chromatin-bound IκB-α interacts with histones H2A and H4 to disrupt the recruitment of the polycomb repressor complex, thereby independently suppressing the transcription of the *HOX* gene [76]. This suggests that these pathways have a regulatory connection that goes beyond conventional NF-κB signaling.

Depending on the situation, HOX proteins can either increase or decrease inflammation, providing possible treatment options [76,85].

### 7.3. HOXA10 and HOXA11 in Pregnancy

*HOXA10* and *HOXA11* are necessary for both implantation and maintenance during pregnancy. Progesterone and estrogen upregulate these genes, while inflammatory mediators such as thrombin and IL-1β suppress them, suggesting that inflammation interferes with *HOX*-mediated pregnancy support. The effects of inflammation and blood loss may be mitigated by restoring *HOX* expression [86].

### 7.4. HOX Genes in Immune Responses

Moreover, *HOX* genes control immune responses. In innate immunity, *HOXA10* induces the E3 ubiquitin ligase Triad1 to end emergency granulopoiesis, while *HOXA9* reverses this effect by suppressing Triad1 [76,87,88].

When T-cells are under nutritional stress, *HOXB9* inhibits NF-κB, which lowers the production of cytokines (such as IFNγ, IL-2, IL-4, and IL-17) and promotes T-cell adaptation [89,90].

Additionally, abnormal T-cell subset imbalances are frequently observed in recurrent spontaneous abortion (RSA) patients, characterized by elevated Th1, Th1/Th2 ratio, and Th17 levels, alongside reduced Th2, Treg, and Treg/Th17 ratios. Notably, Th17 levels are inversely correlated with Treg in these patients, suggesting a disrupted immune tolerance mechanism [90,91]. Given the role of *HOX* genes in modulating immune responses, their potential involvement in regulating T-cell homeostasis in RSA warrants further investigation [91].

## 8. Clinical Implications and Future Directions

### 8.1. Diagnostic Potential of HOX Gene Expression Profiling

Evaluating the expression of the *HOXA10* and *HOXA11* genes has demonstrated tremendous potential as a diagnostic technique for evaluating endometrial receptivity, as they have cycle-dependent expression patterns, and they are an indicator of hormonal responsiveness. These genes provide an essential function in the biochemical and structural changes that occur in the endometrium during the “window of implantation”; therefore, a variety of conditions involving the female reproductive system, which were discussed above, are frequently connected with changes in the expression levels of these *HOX* genes [31,42].

To assess their expression, diverse methods can be used, such as molecular techniques, protein-focused assays, functional studies, in situ techniques, and non-invasive approaches.

Regarding molecular techniques, quantitative PCR (qPCR) is one of the most used techniques for determining the mRNA levels of *HOXA10* and *HOXA11*. The method in question offers great sensitivity and specificity and enables quantitative examination of gene expression. It is commonly utilized for comparing gene expression in various phases of the menstrual cycle and in gynecological conditions such endometriosis and recurrent implantation failure. A highly efficient technique for analyzing global transcriptomes, including the *HOX* genes, is RNA sequencing (RNA-seq). It detects patterns of differential gene expression and related pathways, providing information on the deeper implications of *HOXA10* and *HOXA11* dysregulation. An alternate approach for evaluating mRNA expression is reverse transcription PCR (RT-PCR). It is particularly advantageous in confirming changes in expression levels noticed in larger-scale studies, even though it is not as quantitative as qPCR [92,93].

The *HOXA10* and *HOXA11* proteins are found and measured in endometrial tissues by Western blotting, which validates the functional expression of these genes by providing information involving the protein size and abundance. HOX proteins can also be observed in endometrial tissue sections with immunohistochemistry. This method is particularly advantageous for researching protein expression changes and localization patterns throughout the menstrual cycle. Enzyme-Linked Immunosorbent Assay (ELISA), a quantitative approach for determining the amount of protein in the uterine secretions, can evaluate *HOX* gene-related biomarkers in a non-invasive manner [94].

In situ hybridization provides temporal and spatial data related to gene expression by directly detecting *HOX* mRNA in tissue sections. Higher sensitivity and specificity are provided by more sophisticated versions, such as RNAscope, which make it feasible to identify transcripts with low abundance [92].

Functional studies, such as chromatin immunoprecipitation and luciferase reporter assays, have the examination of transcriptional activity and gene regulation as primary objectives [95,96].

Although they are still in development, liquid biopsy methods endeavor to quantify circulating RNA or proteins associated with the expression of the *HOX* gene, providing a prospective non-invasive diagnostic option [97].

Research has demonstrated that uterine abnormalities like polyps can affect the expression of these genes, which controls processes including endometrial receptivity and ECM remodeling. Molecular screening may offer early diagnostic opportunities for fertility issues associated with these conditions as well as valuable insights into the endometrial environment, but future studies with larger participant numbers and biochemical analysis are required to validate these findings and assess the routine use of these molecular procedures [98].

### 8.2. Therapeutic Approaches Targeting HOX Gene Pathways

New therapies for gynecological disorders that target the expression of the *HOXA10* and *HOXA11* genes concentrate on augmenting endometrial receptivity and maximizing the success of implantation. Among the possible strategies are progesterone modulation, anti-inflammatory therapies, gene editing, and stem cell therapy.

Progesterone, a widely used hormone, regulates *HOXA10* and *HOXA11* expression, indispensable for endometrial receptivity. New available therapies that improve endometrial receptivity may improve fertility by restoring appropriate gene expression [99]. It has also been proven that aromatase inhibitors, which inhibit the enzyme aromatase and hence lower estrogen production, affect reproductive organs in several ways, including altering gene expression. Although there is limited research available regarding how aromatase inhibitors particularly affect *HOXA10* and *HOXA11* expression, studies reveal that fluctuations in estrogen levels triggered by aromatase inhibition might impact the expression of genes linked to endometrial receptivity [100,101,102]. Metformin, an anti-diabetic drug, a member of the biguanide class, has been demonstrated to downregulate the expression of miR-491-3p and miR-1910-3, increasing the expression of *HOX10* and ITGB-3 in the endometrium, as well. The effects were observed, especially, in female patients with PCOS and endometriosis [101,103].

Immunomodulation therapy alters the expression of *HOXA10* and *HOXA11*, which would improve the implantation microenvironment in endometriosis and other inflammatory diseases that interfere with the proper regulation of the *HOXA10* and *HOXA11* genes. A bioactive substance derived from ginseng, protopanaxadiol (PPD), can modify endometrial receptivity and the associated molecular pathways that may involve *HOX* genes, while studies into its direct effect on *HOX* gene expression are still emerging. It has been demonstrated that PPD affects key components that are regulated by the *HOX* genes [104,105].

Moreover, the expression of the *HOXA* genes may be directly modified by technologies, such as CRISPR and RNA interference [105]. By correcting *HOXA10* and *HOXA11* expression deficiencies, these methods may improve endometrial receptivity.

However, stem cell therapy holds significant promise for restoring *HOXA10* and *HOXA11* gene expression in women with EPs, a common cause of impaired endometrial receptivity, because of the compromised *HOXA10* and *HOXA11* expression. Damaged endometrial tissue can be regenerated by making use of stem cells, especially those produced from the endometrium (endometrial stem cells) or pluripotent sources like induced pluripotent stem cells. These cells may divide into decidual, stromal, and epithelial cells, among other cell types seen in the endometrium. Stem cells may restore the appropriate tissue architecture by increasing the uterus’s receptivity for embryo implantation, while modifying the expression of the *HOX* genes [106]. In addition, this therapy has been demonstrated to impact immunological responses and drastically reduce inflammation in the uterus, two aspects that frequently inhibit the implantation process [107,108,109].

### 8.3. Research Gaps and Future Directions

Although the *HOX10* and *HOX11* genes are recognized for regulating endometrial receptivity, research is still being conducted to determine the specific processes causing their dysregulation in EPs. Thus, further research into the molecular, epigenetic, and environmental elements that influence the expression of these genes is required for the development of efficient therapies that would assist the affected patients in restoring their fertility.

First, it is essential to fully understand the signaling pathways that regulate the expression of the *HOX* gene in the endometrium. Understanding how *HOX* gene promoters are dysregulated in polyps will be possible via studying the interactions between these promoters and estrogen, progesterone, and other regulatory factors, such as androgens, vitamin D, and relaxin [31]. Histone modifications and DNA methylation are illustrations of epigenetic alterations that may lead to the overexpression or silencing of the *HOX* gene. The significance of epigenetics in polyps should be investigated further given that it would reveal new treatment targets [110].

However, individuals may be more susceptible to *HOX* gene dysfunction in polyps due to genetic variations. Determining these genetic variants could be useful in predicting their susceptibility and choosing a customized treatment strategy.

Forbye, the endometrial microenvironment, including stromal and epithelial interactions, along with chronic inflammation, could influence *HOX10* and *HOX11* gene expression. Further studies should focus on the interaction between cells and understanding how the ECM remodeling affects *HOX* regulation, as well as being aware of the link between inflammatory cytokines and *HOX* regulation [111].

### 8.4. Need for Longitudinal Studies Examining the Impact of HOX-Targeted Therapies on Fertility Outcomes

The long-term implications on fertility outcomes are still not well defined, despite novel therapies that aim to regulate the expression of the *HOX* gene. This emphasizes the value of long-term research in assessing the safety and effectiveness of *HOX*-targeted treatments. Current studies on *HOX*-targeted treatments frequently include short follow-up periods, small sample sizes, and no control groups. These factors render it challenging to evaluate the long-term effects of treatment and the possibility of unforeseen outcomes such as aberrant endometrial growth or the emergence of malignancies. Furthermore, it proves difficult to generalize results due to disparities in patient groups. For these reasons, longitudinal studies are more than essential to bridge these information gaps, allowing researchers to assess the long-term safety, evaluate the success of implantation, establish the presence of patient subgroups that would benefit from *HOX*-targeted treatments, and profoundly analyze the endometrial environment and its molecular mechanisms.

## 9. Overview of Key Pathways and Genes Influencing *HOX* Function

In this section, we aim to provide a summary of the previous content. To do so, we have created two tables.

The first one outlines the description and impact of key pathways and regulatory mechanisms that influence *HOX* gene function, including Estrogen and Progesterone Signaling, the RA Pathway, the VDR Pathway, TGF-β and Cytokine Networks, MMPs, Epigenetic Modifications, *HOXA10* and *HOXA11* Dysregulation, and NF-κB-*HOX* Crosstalk (see Table 2).

In the second table of this section, we provide a comprehensive overview of *HOXA10* and *HOXA10* and *HOXA11* (joint impact), as well as the NF-κB factor. This overview includes their relation to protein function in signaling, the molecular pathways involved, regulatory factors, associated pathologies, and relevant observations for each case (see Table 3).

## 10. Limitations of the Review

While this review provides valuable insights, several limitations should be acknowledged:Database Selection Bias: The study selection was based on articles retrieved from PubMed, Scopus, and Google Scholar. Relevant studies from other databases or gray literature may have been overlooked, limiting the comprehensiveness of this review.Exclusion Criteria Impact: Strict methodological and reporting criteria may have led to the omission of some relevant studies. Differences in study design, sample sizes, and analytical methods could have affected the generalizability of the findings.Title- and Abstract-Based Screening: The initial screening process was conducted based on article titles and abstracts, which may have led to the exclusion of studies that contained relevant information in the full text but lacked clarity in their abstracts. As a result, some valuable research may have been unintentionally disregarded.Language Restrictions: This review does not explicitly account for language limitations, meaning studies published in languages other than English may have been excluded, potentially reducing the diversity of perspectives.Publication Bias: Studies with significant or positive results are more likely to be published and included, while negative or inconclusive findings might be underrepresented, possibly skewing the overall conclusions.Dependence on Existing Literature: This review relies on previously published studies, meaning any inconsistencies, biases, or gaps in primary research are carried over into the synthesis. This limits the ability to provide entirely objective conclusions.Timeframe Constraints: This review may be restricted to studies published within a specific timeframe, potentially missing older but still relevant research that could provide additional context or alternative perspectives.Heterogeneity of Studies: The included studies may vary significantly in terms of research methods, sample sizes, and analytical approaches, making it challenging to draw uniform conclusions. This variability could introduce inconsistencies in the interpretation of findings.

Despite these limitations, this review followed a systematic selection and evaluation process, ensuring a structured and objective synthesis of existing research. Future studies should aim to address these gaps by incorporating a broader range of sources, refining selection criteria, and standardizing methodologies to improve the reliability and applicability of findings.

## 11. Conclusions

The dysregulation of *HOXA* genes, particularly *HOXA10* and *HOXA11*, plays a crucial role in the pathophysiology of several uterine conditions associated with female infertility, such as EPs, endometriosis, and leiomyomas. In many benign uterine conditions, abnormal *HOXA* gene expression impairs uterine receptivity, contributing to infertility. EPs, often influenced by *HOXA* gene dysregulation, further complicate fertility outcomes by disrupting the endometrial lining and hindering embryo implantation. Moreover, future research is needed regarding the potential implications of *HOXA* gene dysregulation in so-called “recurrent implantation failure” in the absence of any uterine pathology, as these genes are vital for normal endometrial development and implantation.

Understanding the molecular mechanisms behind *HOXA* gene dysfunction, as well as other genes involved in the processes that ensure fertility, offers promising avenues for future research and treatment strategies. Personalized treatment, based on individual genetic profiles, may allow for more effective management of infertility, tailoring therapies to address the root causes rather than symptoms. Additionally, further investigation into *HOXA* gene modulation through pharmacological agents or stem cell therapy could potentially restore uterine receptivity. As research into the role of *HOXA* genes in reproductive health continues to evolve, it is essential to explore the full range of therapeutic options, integrating genetic, hormonal, and environmental factors. The future of fertility treatments lies in personalized, precise approaches that account for the complex genetic and epigenetic factors influencing female infertility.

## Figures and Tables

**Figure 1 biomolecules-15-00563-f001:**
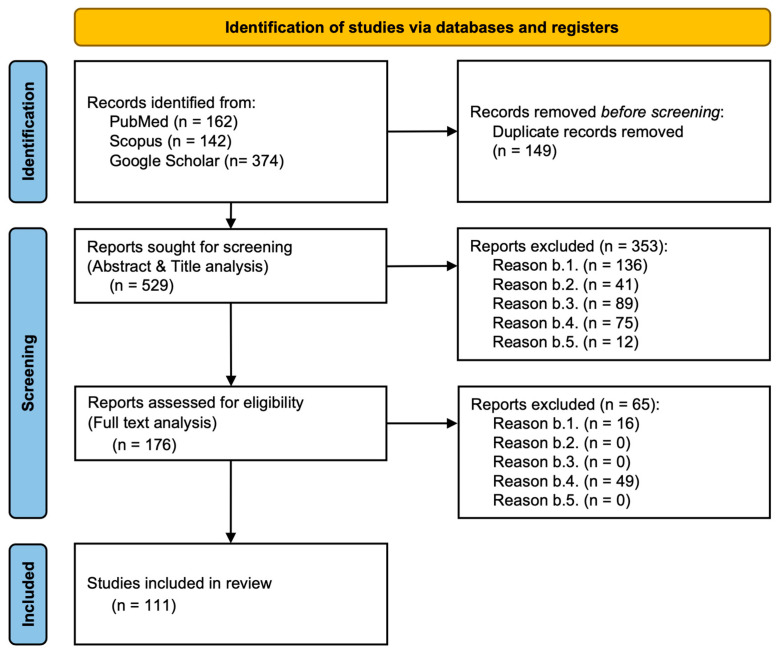
PRISMA flowchart of study selection. The figure illustrates the process of identifying, screening, and selecting relevant studies for inclusion in the review. The flowchart details the number of records retrieved from various databases, the number of duplicate entries removed, and the studies excluded at different screening stages based on predefined exclusion criteria. The final number of studies included in the review is 111. This systematic selection process ensures that the research findings are based on a robust and well-defined dataset.

**Figure 2 biomolecules-15-00563-f002:**
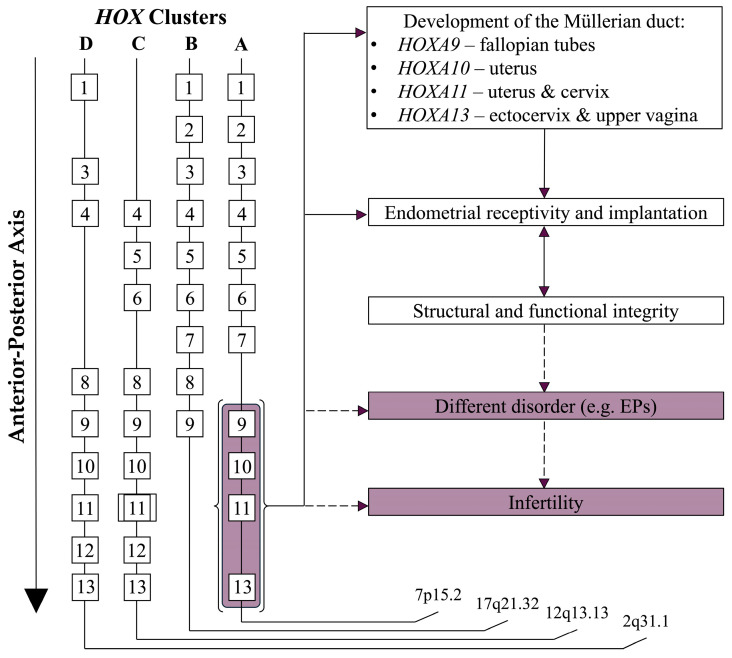
The role of *HOX* genes in female reproductive system development and health. A schematic representation of the spatial expression of *HOX* genes along the anterior–posterior axis of the female reproductive system. The diagram illustrates the roles of individual *HOX* genes, from *HOXA9* in oviduct development to *HOXA13* in ectocervix and upper vagina formation. Abnormal expression patterns are highlighted, showcasing their links to reproductive disorders such as endometrial polyps (EPs). The figure also emphasizes the critical roles of *HOXA10* and *HOXA11* in maintaining endometrial receptivity and supporting embryo implantation.

**Figure 3 biomolecules-15-00563-f003:**
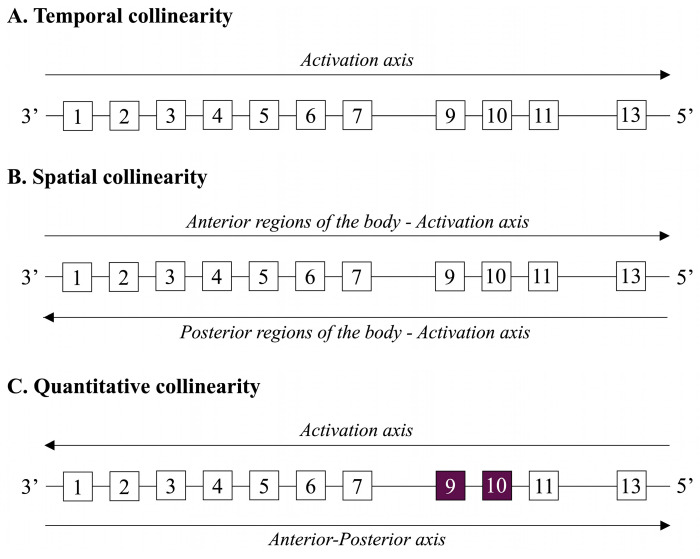
Collinearity in *HOXA* cluster gene expression during embryogenesis. A detailed diagram illustrating the three types of collinearity governing *HOXA* cluster gene expression. (**A**) Temporal collinearity, showing the sequential activation of genes along the cluster from the 3′ end (early in development) to the 5′ end (later in development). (**B**) Spatial collinearity, highlighting the anterior-to-posterior pattern of *HOXA* gene expression, with 3′ genes active in anterior regions and 5′ genes in posterior regions. (**C**). Quantitative collinearity, emphasizing the differential expression levels of *HOXA* genes. Active genes, such as *HOXA9* and *HOXA10*, are marked in dark purple to indicate stronger expression in posterior regions. This section highlights the concept of quantitative collinearity and its potential significance in human development.

**Figure 4 biomolecules-15-00563-f004:**
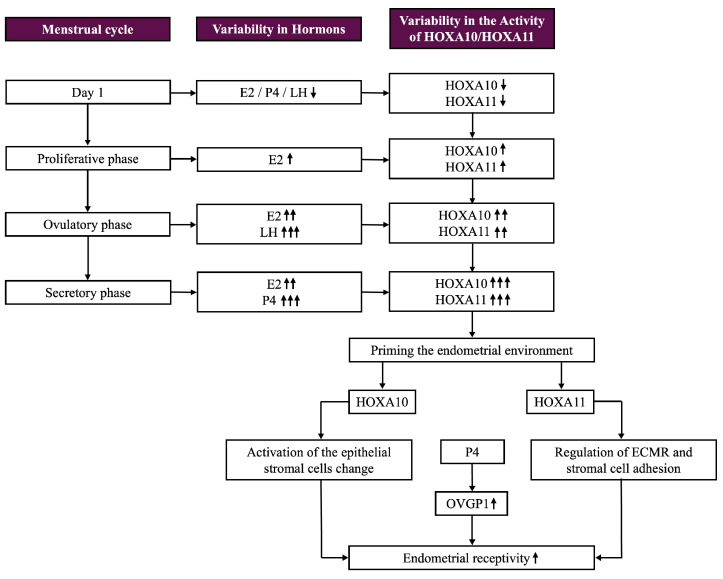
Dynamic *HOXA* gene expression during the menstrual cycle. This flowchart illustrates the regulation and roles of *HOXA10* and *HOXA11* throughout the menstrual cycle phases, highlighting their impact on uterine receptivity. Top section—the hormonal interplay of estradiol (E2), luteinizing hormone (LH), and progesterone (P4) across the proliferative, ovulatory, and secretory phases, driving the sequential upregulation of *HOXA10* and *HOXA11*. Bottom section—the roles of *HOXA10* and *HOXA11* in priming the endometrial environment, such as stromal and epithelial cell activation, extracellular matrix remodeling, and stromal cell adhesion, all contributing to enhanced endometrial receptivity. Note: up arrow—hormones: increased levels; genes: heightened activity. Down arrow—hormones: decreased levels; genes: reduced activity. Multiple arrows indicate greater or lower levels of activity.

**Figure 5 biomolecules-15-00563-f005:**
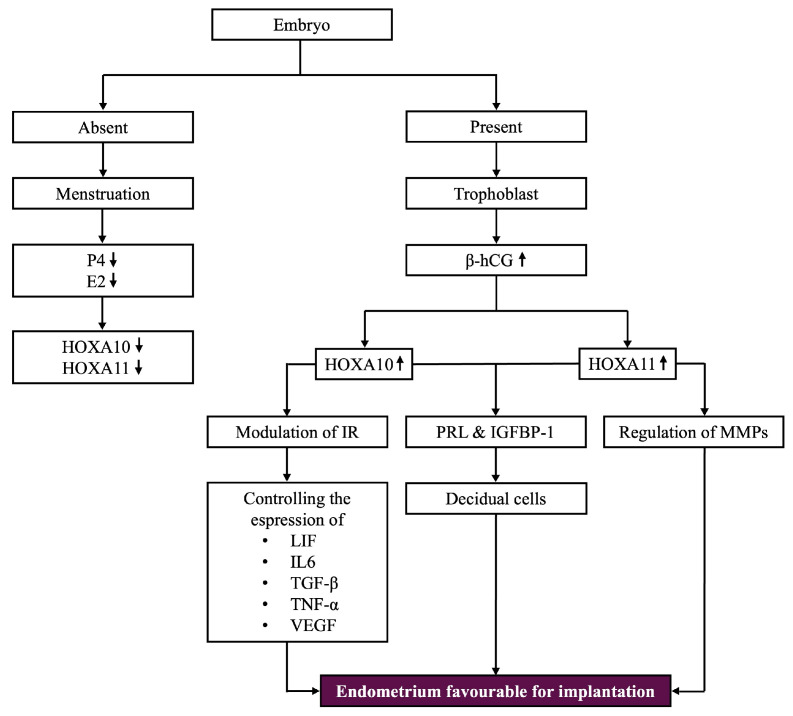
Dynamic *HOXA* gene expression during implantation. This flowchart illustrates the regulation and roles of *HOXA10* and *HOXA11* across the implantation. The contrasting expression patterns in the absence or presence of an embryo. Without pregnancy, decreased P4 and E2 downregulate *HOXA10* and *HOXA11*, leading to menstruation. In pregnancy, trophoblastic β-hCG sustains their expression, promoting decidualization, immunomodulation, and ECM remodeling, which are crucial for implantation. Additional elements include the role of *HOXA* genes in regulating cytokines, PRL, IGFBP-1, and OVGP1 production, emphasizing their broader significance in reproductive success. Note: up arrow—hormones: increased levels; genes: heightened activity. Down arrow—hormones: decreased levels; genes: reduced activity.

**Figure 6 biomolecules-15-00563-f006:**
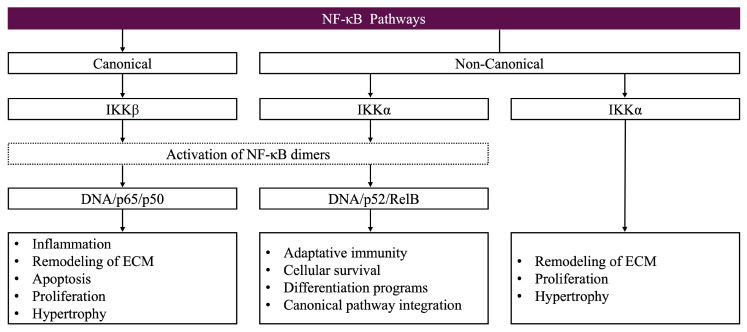
Overview of NF-κB pathways. This figure illustrates the canonical and non-canonical NF-κB signaling pathways, both critical for inflammation, immunity, and development. Canonical pathway: Triggered by stimuli like LPS, TNF-α, and cytokines, the pathway involves IKKβ-mediated phosphorylation of IκB-α. This releases the NF-κB p65/p50 complex, which translocates to the nucleus to regulate genes involved in inflammation, apoptosis, ECM remodeling, and proliferation. Non-canonical pathway: Activated by ligands such as CD40L, BAFF, and RANKL, this pathway utilizes IKKα to phosphorylate p100, converting it into p52. The p52/RelB complex then activates genes essential for adaptive immunity, cellular survival, and differentiation. The pathways exhibit overlapping roles, such as regulating ECM remodeling, cellular hypertrophy, and proliferation, highlighting their integration in immune and developmental processes.

**Table 1 biomolecules-15-00563-t001:** *HOX* gene expression throughout the menstrual cycle.

Phase	*HOXA10/HOXA11* Expression	Regulatory Factors	Impact on Endometrium
Proliferative Phase	Low	Estrogen	Induces growth and proliferation of endometrial lining
Ovulatory Phase	Moderate	Estrogen, LH	Primes the endometrial environment for progesterone responsiveness
Secretory Phase	High	Progesterone, Vitamin D, Relaxin	Supports decidualization, stromal differentiation, and implantation
Post-Implantation	Elevated	Progesterone, β-hCG	Facilitates decidual cell transformation and trophoblast invasion
Menstruation (if no implantation)	Decreased	Regression of Corpus Luteum	Leads to shedding of functional endometrial layer

**Table 2 biomolecules-15-00563-t002:** Key pathways and regulatory mechanisms influencing *HOX* gene function.

Pathway/Regulatory Factor	Description	Impact
Estrogen and Progesterone Signaling	Works synergistically to modulate *HOXA10/HOXA11* expression during menstrual phases, peaking in the secretory phase to prepare for implantation.	Key drivers of uterine receptivity, particularly during the secretory phase, ensuring proper endometrial preparation for implantation.
RA Pathway	Influences 3′ *HOX* gene expression, including *HOXA9*, critical for Müllerian duct development.	Regulates the development of the anterior–posterior axis and ensures proper organ formation along the reproductive tract.
VDR Pathway	Upregulates *HOXA10* and *HOXA11*, enhancing uterine receptivity by modifying stromal cell adhesion and immune environment.	Enhances stromal cell adhesion and supports immune modulation to create a favorable implantation environment.
TGF-β and Cytokine Networks	TGF-β and pro-inflammatory cytokines like IL-6 and TNF-α regulate cellular differentiation, with disruption leading to infertility and endometriosis.	Critical in cellular differentiation and immune balance; disruption causes inflammation, tissue damage, and reduced reproductive capability.
MMPs	*HOX* genes, particularly *HOXA10* and *HOXA11*, regulate MMPs critical for ECM remodeling during trophoblast invasion.	Essential for ECM remodeling, enabling embryo invasion into the endometrial lining during implantation.
Epigenetic Modifications	DNA methylation of *HOXA10/HOXA11* promoters in pathological states (e.g., endometriosis, adenomyosis) results in decreased gene expression and altered endometrial receptivity.	Suppresses *HOX* gene expression, compromising decidualization and implantation success.
*HOXA10* and *HOXA11* Dysregulation	Lower expression in endometriosis reduces implantation success, while hypermethylation in adenomyosis affects decidualization and pregnancy outcomes.	Causes implantation failure and adverse pregnancy outcomes; central to many gynecological pathologies.
NF-κB-*HOX* Crosstalk	NF-κB pathways regulate inflammation and immunity, interacting with *HOX* genes to modulate embryonic patterning and immune responses.	Highlights the interconnectedness of inflammatory responses and developmental processes, with implications for reproductive disorders and systemic immune function.

**Table 3 biomolecules-15-00563-t003:** Comprehensive overview of *HOX* genes, associated pathways, and pathological implications.

Gene/Factor	Protein Function in Signaling	Molecular Pathway(s) Involved	Regulatory Factors	Associated Pathologies	Observations
*HOXA10*	Facilitates stromal decidualization and regulates ECM remodeling	MMPs, LIF, VEGF signaling	Progesterone, estrogen, vitamin D, relaxin	Infertility, endometriosis, adenomyosis, EPs	Vitamin D increases *HOXA10* expression, enhancing endometrial receptivity. Dysregulation leads to failed implantation
*HOXA10* and *HOXA11* (joint impact)	Orchestrates implantation and immune microenvironment	ECM remodeling, cytokine network (IL-6, LIF, TGF-β)	Progesterone, estrogen, vitamin D	Adenomyosis, endometriosis, infertility	Coordinate stromal and epithelial adhesion during implantation
NF-κB	Mediates inflammatory response and embryonic development	Canonical and non-canonical NF-κB pathways	Inflammatory cytokines (e.g., TNF-α, IL-1β), lipopolysaccharide (LPS)	Obesity, inflammatory diseases, atherosclerosis	Regulates dorsal/ventral patterning alongside *HOX* genes; interacts with *HOXA10* during endometrial remodeling

## Data Availability

No new data were created.

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
