# Peer review of "HOXA10 and HOXA11 in Human Endometrial Benign Disorders: Unraveling Molecular Pathways and Their Impact on Reproduction"

_biomolecules, 2025, doi:10.3390/biom15040563_

Round 1

Reviewer 1 Report

Comments and Suggestions for Authors

Your paper untitled "HOXA10 and HOXA11 in Human Endometrial Benign Disorders: Unraveling Molecular Pathways and Their Impact on Reproduction" from Pirlog et al.  is good but it is also the nth review on this subject in gynaecology and reproductive medicine. It adds nothing to what we already know. However, it is well written and well documented, with didactic diagrams. Many mentioned markers were involved in a lot of gynecological pathologies. The role of HOXA 10 and 11 has been underscored in the past but, unfortunately, other markers were mentioned by several authors, particularly in reproductive medicine  without changing our results. Embryo implantation is a multifactorial event including the role of the embryo itself and the precise dialog utero- fetal is not yet clearly understood.

Author Response

1. Summary

Thank you very much for taking the time to review this manuscript. Please find the detailed responses below and the corresponding revisions and corrections highlighted in the re-submitted files.

2. Point-by-point response to Comments and Suggestions for Authors

Comment #1. Your paper untitled "HOXA10 and HOXA11 in Human Endometrial Benign Disorders: Unraveling Molecular Pathways and Their Impact on Reproduction" from Pirlog et al.  is good but it is also the nth review on this subject in gynaecology and reproductive medicine. It adds nothing to what we already know. However, it is well written and well documented, with didactic diagrams. Many mentioned markers were involved in a lot of gynecological pathologies. The role of HOXA 10 and 11 has been underscored in the past but, unfortunately, other markers were mentioned by several authors, particularly in reproductive medicine  without changing our results. Embryo implantation is a multifactorial event including the role of the embryo itself and the precise dialog utero- fetal is not yet clearly understood.

Response #1. We would like to sincerely thank the reviewer for taking the time to evaluate our manuscript and for the thoughtful comments. We appreciate the acknowledgement of the manuscript’s clarity, documentation, and the quality of the diagrams. We fully agree that embryo implantation is a highly complex and multifactorial process, involving not only endometrial factors such as HOXA10 and HOXA11, but also critical contributions from the embryo and the dynamic utero-embryonic dialogue, which remains incompletely understood.

Although the role of HOXA10 and HOXA11 has indeed been explored previously, our aim was to provide a focused and integrative overview of their expression in benign endometrial conditions (including endometriosis, polyps, and hyperplasia) and their specific implications for endometrial receptivity and infertility. By compiling and contextualizing current findings, we hoped to highlight knowledge gaps and support the need for further research into molecular diagnostics and therapeutic targets in clinical reproductive medicine.

We value the reviewer’s perspective and believe the feedback has helped us reflect on the scope and clarity of our work.

Reviewer 2 Report

Comments and Suggestions for Authors

Comments to the Author
This article mainly reviews the roles of homeobox genes (HOXA10 and HOXA11) in reproductive health and their dysregulation in benign pathologies associated with infertility, such as endometriosis, adenomyosis, and endometrial polyps. The information reviewed in this article would be helpful for further research work about human reproductive disorders. So, I recommend the following minor revision points to improve the quality of this review article.

Minor points
1. The authors should provide proper keywords.
2. In the introduction, the authors should give some background to explain why the dysregulation of HOXA10/HOXA11 genes are important to development of endometriosis and infertility.
3. The authors should check the format type, especially some technical words are typed in uppercase, and some are typed in lowercase letters.
4. Figure 7, please give proper subtitles.

Author Response

1. Summary

Thank you very much for taking the time to review this manuscript. Please find the detailed responses below and the corresponding revisions and corrections highlighted in the re-submitted files.

2. Point-by-point response to Comments and Suggestions for Authors

Comment #1. This article mainly reviews the roles of homeobox genes (HOXA10 and HOXA11) in reproductive health and their dysregulation in benign pathologies associated with infertility, such as endometriosis, adenomyosis, and endometrial polyps. The information reviewed in this article would be helpful for further research work about human reproductive disorders.

Response #1. We sincerely thank the reviewer for the positive and encouraging feedback. We are pleased that the relevance and potential usefulness of our review in the context of ongoing and future research on human reproductive disorders was recognized. Our goal was to synthesize and critically present current knowledge on HOXA10 and HOXA11 in benign endometrial pathologies, with a focus on their regulatory roles in endometrial receptivity and infertility. We hope that this work contributes to advancing molecular insight and supporting further investigations in this complex and evolving field.

--

Comment #2. The authors should provide proper keywords.

Response #2. We thank the reviewer for this helpful suggestion. In response, we have revised the list of keywords to better reflect the core focus and content of the manuscript. The updated keywords emphasize key concepts such as HOXA10, HOXA11, endometrial receptivity, infertility, and specific benign endometrial pathologies, aiming to improve the article’s discoverability and alignment with its scientific scope.

--

Comment #3. In the introduction, the authors should give some background to explain why the dysregulation of HOXA10/HOXA11 genes are important to development of endometriosis and infertility.

Response #3. We thank the reviewer for this valuable suggestion. In response, we have revised the Introduction to provide clearer background information on the importance of HOXA10 and HOXA11 dysregulation in the context of endometriosis and infertility. Specifically, we have added a new paragraph that highlights the multifactorial nature of endometriosis-associated infertility and explains how the downregulation of HOXA10 and HOXA11 contributes to impaired decidualization, immune dysfunction, and reduced endometrial receptivity. This addition clarifies the molecular link between HOX gene dysregulation and implantation failure, even in the absence of anatomical abnormalities. We also acknowledge that the molecular mechanisms underlying endometrial function, receptivity, and implantation are highly complex and remain incompletely understood. The aim of this review was to synthesize current findings and emphasize the specific roles of HOXA10 and HOXA11 in these processes, particularly in the context of benign endometrial pathologies and reproductive failure. While we discussed in dedicated sections of the manuscript the interconnections between these genes and endometriosis in detail, we initially chose to keep the Introduction concise and focused, in line with the purpose of this section. However, following the reviewer’s suggestion, we agree that a brief contextualization was beneficial, and we have now incorporated it accordingly. (lines 54-66)

--

Comment #4. The authors should check the format type, especially some technical words are typed in uppercase, and some are typed in lowercase letters.

Response #4. We thank the reviewer for bringing this to our attention. In response, we carefully reviewed the manuscript and performed a detailed analysis of the formatting and capitalization of technical terms. We identified several inconsistencies, particularly in the use of gene and protein names (e.g., HOXA10, HOXA11) and abbreviations such as P4 and E2. These have now been corrected according to accepted scientific conventions: gene names are italicized, protein names are capitalized but not italicized, and hormone abbreviations are consistently used and defined upon first mention. We appreciate the reviewer’s comment, which helped us improve the clarity and precision of our manuscript.

--

Comment #5. Figure 7, please give proper subtitles.

Response #5. We sincerely thank the reviewer and the editorial team for bringing this to our attention. We apologize for the oversight in submitting the same image as both Figure 7 in the manuscript and the graphical abstract, which does not comply with the journal’s submission guidelines. In response, we have removed Figure 7 from the manuscript to avoid redundancy and ensure proper alignment with the graphical abstract policy. We appreciate your guidance and understanding.

Reviewer 3 Report

Comments and Suggestions for Authors

 This review highlights the molecular roles of HOXA10/HOXA11 genes as biomarkers and therapeutic goals to enhance fertility outcomes and define reproductive pathologies. The review is highly interesting and informative. I didn’t find any problem to comment on. The review in its current form is deserved to be accepted for publication.

  • Key words should be rearranged alphabetically and should be differed from those mentioned in the title to expand the visibility in the research engine.

Author Response

1. Summary

Thank you very much for taking the time to review this manuscript. Please find the detailed responses below and the corresponding revisions and corrections highlighted in the re-submitted files.

2. Point-by-point response to Comments and Suggestions for Authors

Comment #1. This review highlights the molecular roles of HOXA10/HOXA11 genes as biomarkers and therapeutic goals to enhance fertility outcomes and define reproductive pathologies. The review is highly interesting and informative. I didn’t find any problem to comment on. The review in its current form is deserved to be accepted for publication.

Response #1. We sincerely thank the reviewer for the positive and encouraging feedback. We are pleased to know that the review was found to be informative and of interest. We truly appreciate your supportive remarks and are grateful for your recommendation for publication.

---------

Comment #2. Key words should be rearranged alphabetically and should be differed from those mentioned in the title to expand the visibility in the research engine.

Response #2. We thank the reviewer for this helpful and constructive suggestion. In response, we have revised the list of keywords to better reflect the main themes of the manuscript while avoiding repetition of terms already included in the title. The updated keywords are now arranged alphabetically and emphasize important concepts such as endometrial receptivity, epigenetic regulation, implantation failure, and specific benign endometrial pathologies. These changes aim to enhance the manuscript’s discoverability and visibility in academic search engines and indexing databases. (Lines 36-38)